# Resolving multi-image spatial lipidomic responses to inhaled toxicants by machine learning

Nathanial C. Stevens [1], Tong Shen [1], Joshua Martinez[2], Veneese J. B. Evans[2], Morgan C. Domanico [2], Elizabeth K. Neumann[3], Laura S. Van Winkle[2,4] & Oliver Fiehn [1] ✉

Regional responses to inhaled toxicants are essential to understand the pathogenesis of lung disease under exposure to air pollution. We evaluate the effect of combined allergen sensitization and ozone exposure on eliciting spatial differences in lipid distribution in the mouse lung that may contribute to ozone-induced exacerbations in asthma. We demonstrate the ability to normalize and segment high resolution mass spectrometry imaging data by applying established machine learning algorithms. Interestingly, our segmented regions overlap with histologically validated lung regions, enabling regional analysis across biological replicates. Our data reveal differences in the abundance of spatially distinct lipids, support the potential role of lipid saturation in healthy lung function, and highlight sex differences in regional lung lipid distribution following ozone exposure. Our study provides a framework for future mass spectrometry imaging experiments capable of relative quantification across biological replicates and expansion to multiple sample types, including human tissue.

More than 137 million people in the United States live in areas with unhealthy levels of air pollution[1]. Exposure to major components of air pollution, including particulate matter and oxidant gases, are well characterized for their ability to worsen existing lung disease and to potentially cause new-onset respiratory disease[2–4]. Despite extensive evaluation of the acute and chronic adverse health outcomes of inhaled toxicants, the molecular mechanisms underlying these effects are still not well understood. Importantly, previous studies have demonstrated that particulate matter and oxidant gases such as ozone ($O_3$) elicit site-specific toxicity, which is dependent upon the physiochemical properties of a toxicant and its inhaled concentration[5–7]. Notably, the region-specific effects of $O_3$ exposure on the conducting airways are well studied, which acutely induces airway hyperreactivity, airway inflammation, and damages lung surfactant. $O_3$ exposure is also a well-known risk factor for exacerbating pre-existing asthma in

humans, although potential mechanisms that may explain this association are not well understood[1,4,5]. The use of combined exposure models incorporating $O_3$ and common human allergens such as house dust mite (HDM) may elucidate mechanisms of $O_3$-induced exacerbations in asthma.

In addition to the region-specific effects of an inhaled toxicant, the wide array of cell types within the lung, differences in xenobiotic metabolism, and unequal distribution of cell populations along the respiratory tract all lead to effects that are often confined to individual cell types or lung regions[8,9]. Elucidating site-specific responses is therefore necessary for implicating individual types of cells or regions in promoting lung disease and to develop targeted therapeutic approaches to mitigate the outcomes of inhaled toxicant exposure. Prior studies evaluating regional differences within the lung have implemented techniques such as gross lung microdissection, which

[1]Genome Center, University of California Davis, Davis, CA, USA. [2]Center for Health and the Environment, University of California Davis, Davis, CA, USA. [3]Department of Chemistry, University of California Davis, Davis, CA, USA. [4]Department of Anatomy, Physiology and Cell Biology School of Veterinary Medicine, University of California Davis, Davis, CA, USA. ✉e-mail: ofiehn@ucdavis.edu

isolates major lung regions for further processing and downstream analysis[10]. These studies have illustrated notable differences in xenobiotic metabolism, gene expression, and global metabolite abundance while demonstrating how inherent differences between regions are modulated by toxicant exposure[11–14]. However, analysis of microdissected tissues does not isolate the contributions of individual cell types that influence responses to inhaled toxicant exposure, despite the clear advantages of this approach over whole lung analysis in terms of spatial resolution. Emerging techniques such as single-cell RNA sequencing (scRNA-Seq) and spatial transcriptomics may circumvent some limitations associated with lung microdissection by simultaneously identifying changes in gene expression and tracing these changes back to individual types of cells[9,15]. Conversely, high spatial resolution metabolomics analysis in the lung remains a significant challenge due to difficulties stemming from throughput of existing analytical methods and lung-specific issues regarding compatible sample preparation methods for spatial metabolomics analysis[16,17]. Nonetheless, high resolution spatial metabolomics studies are needed to contextualize the functional changes downstream of gene expression within the lung following inhaled toxicant exposure.

Mass spectrometry imaging (MSI) demonstrates the potential to facilitate high spatial resolution metabolomics analysis. MSI has recently been utilized in a variety of applications to assess the localization of lipids and metabolites at near single-cell resolution in the lung and at single cell resolution in other tissue types[18–22]. Despite technical advancements in MSI data acquisition, processing and analysis of MSI data specifically for large-scale metabolomics or lipidomics studies remains challenging. Both commercial and open-source software have been developed to perform common MSI data processing tasks and are continually advancing in their capability and functionality[23–27]. Nonetheless, the burgeoning popularity of spatial metabolomics studies and the relatively low availability of tools to probe MSI datasets compared to traditional liquid chromatography tandem mass spectrometry (LC-MS/MS) untargeted metabolomics underscores the need for additional resources, specifically software capable of performing MSI data analysis across biological replicates.

Including biological replicates is recognized as necessary for expanding the scope of MSI studies. Indeed, recent applications of MSI have incorporated replicates and qualitative assessment of metabolite and lipid spatial distribution to drive biologically meaningful conclusions from MSI data[28–30]. However, the ability to conduct statistical analysis across biological replicates in MSI experiments is currently limited. Likewise, applying previously developed segmentation methods to lung tissue is especially difficult due to morphological features exclusive to the lung compared to common tissue types analyzed by MSI such as brain or kidney tissue.

Our present study addresses these gaps in MSI studies of lung tissue by providing comprehensive, versatile analysis scripts to compare biological replicates across study groups. Secondly, we apply our MSI analysis workflow to determine the effects of acute $O_3$ exposure on lung lipids in a mouse model of allergic asthma at high spatial resolution in morphologically relevant lung regions. This analysis builds on our previous study demonstrating significant changes in sphingolipid and glycerophospholipid abundance of microdissected airways following combined $O_3$ exposure and allergic sensitization to HDM by localizing specific changes in whole lung tissues[12]. Lastly, the comparisons drawn between our previous study and the results of our MSI study emphasize the detailed spatial information that can be obtained from MSI compared to LC-MS/MS of bulk dissected tissue.

## Results

### Histological analysis of mouse lungs

We have previously reported the effects of combined HDM/ozone exposure in mice on lung physiology and lipidomic profiles of microdissected lung airways and parenchyma[12]. Our previous findings demonstrated a synergistic increase in airway hyperreactivity and airway inflammation in male mice that was not observed in female mice. Due to the relatively modest effects of either HDM or ozone alone in significantly altering global lipid abundance within microdissected tissues, our present MSI study focused solely on the effects of combined HDM/ozone exposure. Therefore, lungs for six mice with HDM/ozone exposure were compared to six control lungs, three per sex in each group. We here confirmed major structural changes in HDM/ozone exposed lungs compared to control lungs using H&E staining (Supplementary Fig. 2). Specifically, combined HDM/ozone resulted in increased airway inflammation in males compared to females, which was evidenced by thickening of the airway epithelium relative to control treated mice. H&E staining of agarose-inflated lung sections in general was less informative for obtaining detailed morphological insights compared to our previous study analyzing paraformaldehyde fixed and paraffin embedded H&E sections. However, H&E-stained sections were still suitable to confirm location specific features corresponding to our MSI data.

### Summary of annotated lipids

For each lung section of approximately 0.2 cm$^2$, positive and negative ion mode mass spectrometry imaging was conducted using a 10 μm raster on sections, yielding MSI files of up to 10 GB. We initially exported all raw data files into SCiLS software for conversion into imzML format. All processing and analysis of the exported imzML files was completed in R v.4.3.3 using comprehensive analysis scripts utilizing functions from over 15 separate R packages, which we developed into a standalone R package called RegioMSI. Peak detection, binning, and alignment were completed by the Cardinal R package, which allows all peaks across multiple experimental runs to be processed simultaneously[31]. By concurrent processing of all samples, we obtained a single alignment list of approximately 1100 peaks for 12 positive mode sections and 700 peaks for 12 negative mode sections, after binning profile spectra for subsequent peak annotation (Supplementary Data 3).

We assigned annotations to detected peaks by untargeted LC-MS/MS-based lipidomics validation of tissues scraped from ITO slides following MSI data acquisition or by matching m/z values to our previous untargeted lipidomics data of microdissection lung tissue treated under identical experimental conditions[12]. We did not evaluate the effect of matrix application on lipid detection in LC-MS/MS, although previous reports suggest that the number of lipids identified by LC-MS/MS is not significantly affected by matrix deposition[32]. Matching m/z values were determined using a 10 mDa mass error, and we excluded matches to LC-MS/MS data with a low likelihood of detection in a specific MSI ionization mode[33]. Specifically, we excluded features in positive ionization mode MSI that matched phosphatidylinositol and phosphatidylserine species detected in LC-MS/MS data. Additionally, we manually removed m/z values representing noisy or background ion images to improve downstream image segmentation and statistical analysis. Following these filtering steps, 119 peaks detected in positive ionization mode and 83 peaks detected in negative ionization mode were annotated to each LC-MS/MS data set (Supplementary Data 1). Over 75% of annotated lipids were glycerophospholipids between 700 and 900 m/z, whereas sphingolipids and other major lipid classes comprised less than 25% of 202 total annotated lipids across both ionization modes (Fig. 1b, Supplementary Data 2).

### Intra- and inter-sample normalization by sparse LOESS

We next normalized the signal intensities for all annotated compounds in each tissue section to mitigate sources of technical variance. Marked intrasample signal drift in the raw intensity values was observed in both positive and negative mode, which was also reflected in raw total ion count ion images prior to normalization (Fig. 2a, Supplementary

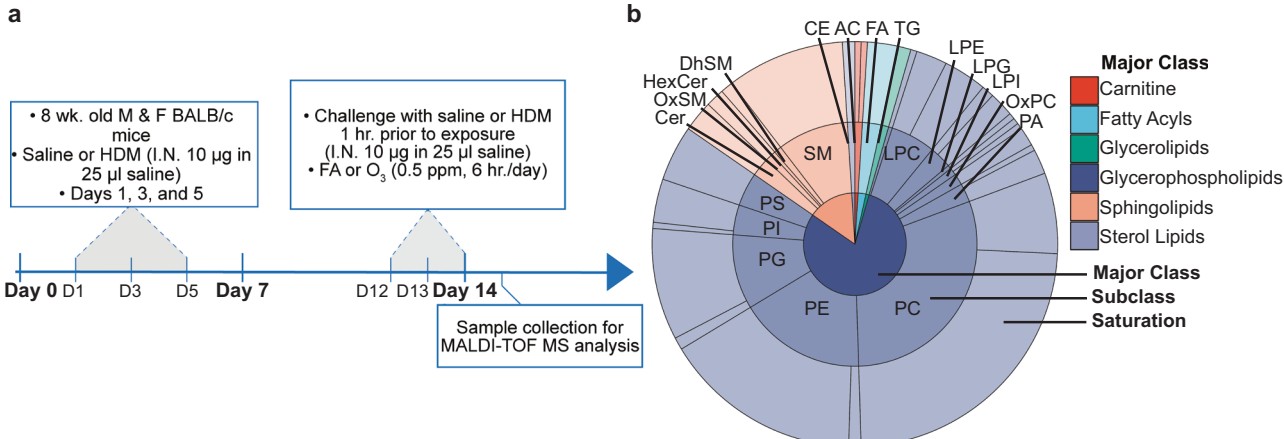

**Fig. 1 | Experimental design and summary of annotated lipids across positive and negative ionization modes. a** Experimental design including allergic sensitization, challenge, and ozone exposure. Left lung lobes were collected 24 h following the final day of ozone exposure. **b** Summary of all lipid annotations grouped by major class, subclass, and saturation validated by LC-MS/MS of scraped tissue slides and microdissected lung tissue under identical experimental conditions as previously reported[12]. The individual annotation list is included in Supplementary Data 1 and a supplemental list containing the proportion of lipids in each major class, including saturated and unsaturated lipid species, is included in Supplementary Data 2. Source data for panel b are provided as a Source Data file.

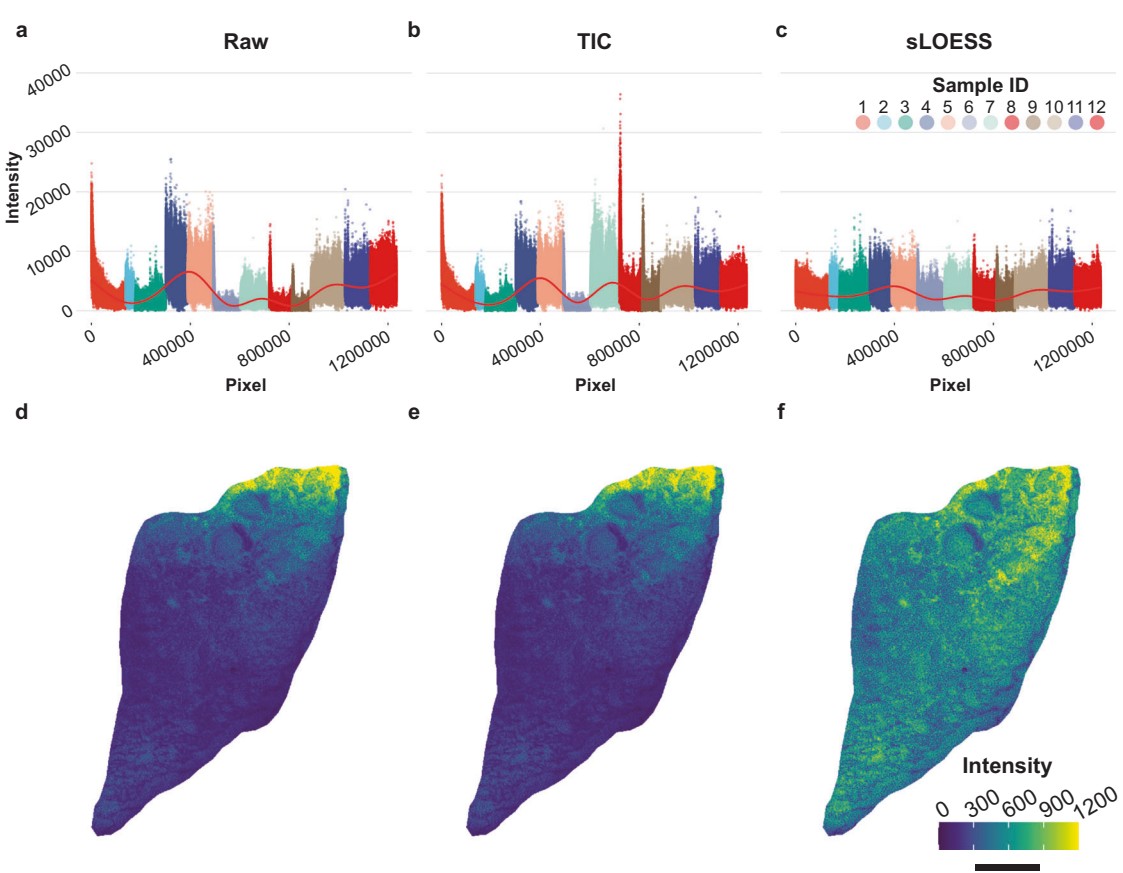

**Fig. 2 | Effect of TIC and sparse LOESS normalization on technical variance in MSI signal intensity.** Total ion current scatterplots of signal intensity vs. pixel number according to acquisition order in negative ionization mode for (**a**) raw, (**b**) TIC, and (**c**) sparse LOESS normalized data. **d**–**f** Total ion current ion images for a representative sample displaying the raw, TIC, and sparse LOESS normalized signal intensity in negative mode, respectively. TIC values correspond to the sum intensity of all annotated compounds for each pixel. Sample ID reflects the identity of individual imaging runs in order of data acquisition. Bar = 1 mm. Source data for panels (**a**–**c**) are provided as Source Data files.

Fig. 3, Fig. 2d). Similarly, raw intersample signal intensity ranges varied by more than 50% across several samples regardless of ionization mode (Fig. 2a, Supplementary Fig. 3). Notably, routine total ion current (TIC) normalization partially corrected intrasample signal intensities in positive and negative mode but unsuccessfully reduced variance between samples (Fig. 2b, Supplementary Fig. 3).

Therefore, we applied LOESS across all samples to minimize technical bias that would otherwise limit downstream statistical comparisons across biological replicates and treatment groups. LOESS normalization was confined to every non-zero intensity pixel to correct signal drift while preserving the spatial distributions of each annotated compound. Furthermore, we used a variable LOESS span that factored approximately 10% of the total pixel number per sample into the LOESS algorithm to prevent over smoothing. Sparse LOESS normalization successfully corrected intrasample signal drift and greatly reduced signal variability across both positive and negative ionization mode samples compared to both raw and TIC normalized data (Fig. 2c, Supplementary Fig. 3). The effect of sparse LOESS normalization in reducing intrasample signal drift was especially prominent in comparing sparse LOESS normalized ion images to either raw or TIC normalized ion images displaying pixel TIC intensities (Fig. 2d–f).

## Assessment of spatial differences among annotated lipids

Visualizing ion images normalized by sparse LOESS enabled us to determine differences in lipid spatial distribution and localization while minimizing technical artifacts impacting comparisons across samples. We observed many phospholipids confined to specific section areas, which was consistent across all samples in positive and negative ionization modes. For example, MSI features annotated to phosphatidylethanolamine (PE) 18:0/22:6, phosphatidylinositol (PI) 36:4, and phosphatidic acid (PA) 16:0/16:0 in negative ionization mode were highly localized to specific regions within lung tissues (Fig. 3a–c, Supplementary Fig. 4). Our data supported the notion that PA 16:0/16:0 was likely formed as an in-source fragment of PC 32:0. Importantly, these three lipids exhibited distinct localization where the greatest abundance of one lipid did not overlap with another and displayed distributions that appeared to be independent of potential effects induced by treatment or sex.

We also observed differences in lipid species localization in positive ionization mode (Supplementary Fig. 5). However, inconsistent tissue section morphology resulted in patterns of lipid spatial distribution that were difficult to distinguish by manual inspection of ion images alone. Specifically, not all sections analyzed in positive mode contained prominent morphological features such as proximal airways identifiable by visualizing individual ion images for annotated lipids (Supplementary Fig. 5). Consequently, we were unable to uniformly define regions of interest across all samples in both ionization modes by individual lipid species distribution.

## Image segmentation and colocalization

To address inconsistencies in section morphology preventing robust region of interest selection and subsequent statistical analysis across treatment groups, we performed unsupervised machine learning-based image segmentation based on all annotated lipids in each sample. We adapted the KNN algorithm and graph-based clustering approach used by Seurat, a commonly used R package for scRNA-Seq analysis, to perform segmentation of each lung tissue section in both ionization modes[34]. This segmentation approach was chosen over spatial shrunken centroids (SSC) classification used by both commercial and Cardinal software because the default implementation of SSC by both software tools performs segmentation with limited flexibility for selecting only annotated features and parameter optimization compared to Seurat[24]. Unsupervised image segmentation resulted in distinct clustering of multiple regions in each tissue section (Fig. 4a). Visualizing identified clusters allowed us to accurately assign grouped pixels to H&E validated morphological regions of interest while isolating clusters likely arising from technical artifacts in data acquisition (Fig. 4b–d, Supplementary Fig. 2, Supplementary Fig. 6). Airway and alveolar epithelium were distinguished by unsupervised segmentation in all sections analyzed across both ionization modes. However, our unsupervised approach was unable to completely exclude some pixels that did not overlap with a morphological region of interest (Fig. 4b–d). Nonetheless, the proportion of these pixels relative to the total pixels contained within each cluster did not prevent us from selecting regions of interest from our segmentation results.

Next, we assessed the colocalization of individual lipid species in each segmented cluster while focusing our analysis on clusters overlapping with morphological regions of interest. We observed an interesting relationship between the degree of lipid saturation, class, and region. Localized in the airway epithelium, polyunsaturated phosphatidylcholines (PC) and sphingomyelins (SM) 44:1 and 44:2 were consistently among the top-5 most abundant lipids. In positive ionization mode, PCs containing four or fewer double bonds, as well as lyso-PC and lyso-PE defined the alveolar epithelium (Fig. 4b, d, e, Supplementary Data 4). Conversely, the top-5 lipids detected in each region among negative mode samples did not display a pattern of decreasing saturation degree from proximal to distal lung regions. Each region was instead distinguished by colocalization of specific lipid classes. Polyunsaturated fatty acids such as docosahexaenoic acid (FA 22:6) and polyunsaturated PE were detected primarily in the airway epithelium, whereas both saturated and unsaturated phosphatidylglycerols (PG) predominated the alveolar epithelium (Supplementary Data 4).

Finally, unsupervised segmentation of specific lung sections based on MSI data enabled a detailed approximation of lipid abundance in individual lung regions not included in downstream statistical analyses. We excluded specific regions from comparisons across treatment groups since regions such as the airway basement membrane and distal airway epithelium were not clearly identified in all lung sections. Nonetheless, our segmentation approach discerned adjacent lung regions with high spatial resolution, which was supported by our histological data (Fig. 5a, b). Specifically, our segmentation approach resolved the airway epithelium from the basement membrane in specific samples (Fig. 5a, b). This finding was supported by visualizing the abundance distributions of two polyunsaturated PCs corresponding to each of these regions, which overlapped with both our H&E data and segmentation results (Fig. 5c, d). These results demonstrate the capability of our analysis to segment the lung into distinct regions based solely upon localization of annotated lipids and to measure regional lipid abundance at high spatial resolution.

## Local effects of ozone/HDM exposure on lipid composition

Lastly, the segmented images of each sample were used to compare the effects of combined HDM + $O_3$ exposure between lung regions in both male and female mice. Prior to statistical analysis, we log-transformed the average abundances of each lipid per pixel to obtain a distribution that was approximately normal. We initially conducted enrichment analysis to assess differences in abundance among lipids belonging to structurally similar classes[35]. We observed significant decreases in multiple unsaturated lipid classes in the female HDM + $O_3$ group relative to control in both the airway and alveolar epithelium, including unsaturated PE and PS (Fig. 6a, c). However, unsaturated FA, PG, and PI in addition to saturated PG, PE, and LPE were decreased in the alveolar epithelium but not in the airway epithelium (Fig. 6c). These multivariate changes were followed by univariate statistical analysis, which identified individual lipid species decreased by HDM + $O_3$ (Fig. 6b, d, Supplementary Data 5). Interestingly, we observed a significant decrease in individual SM species and an increase in ceramide 34:1 exclusive to the airway epithelium, which were not reflected by class-based enrichment analysis (Fig. 6a, b).

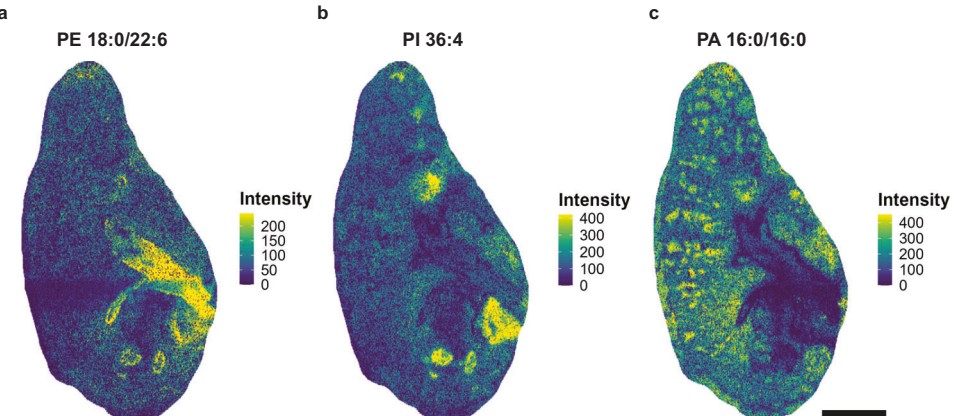

**Fig. 3 | Spatial distribution of individual phospholipid species in sparse LOESS-normalized ion images.** Ion images representing pixel intensities of annotated phospholipids in negative mode, including (**a**) PE 18:0/22:6, (**b**) PI 36:4, and (**c**) PA 16:0/16:0. Ion images correspond to a representative sample from the female control group. Ion images for each lipid species from each individual sample is included in Supplementary Fig. 4. Bar = 1 mm. No acyl chain information was available for PI 36:4 based on the LC-MS/MS reference library used for peak annotation.

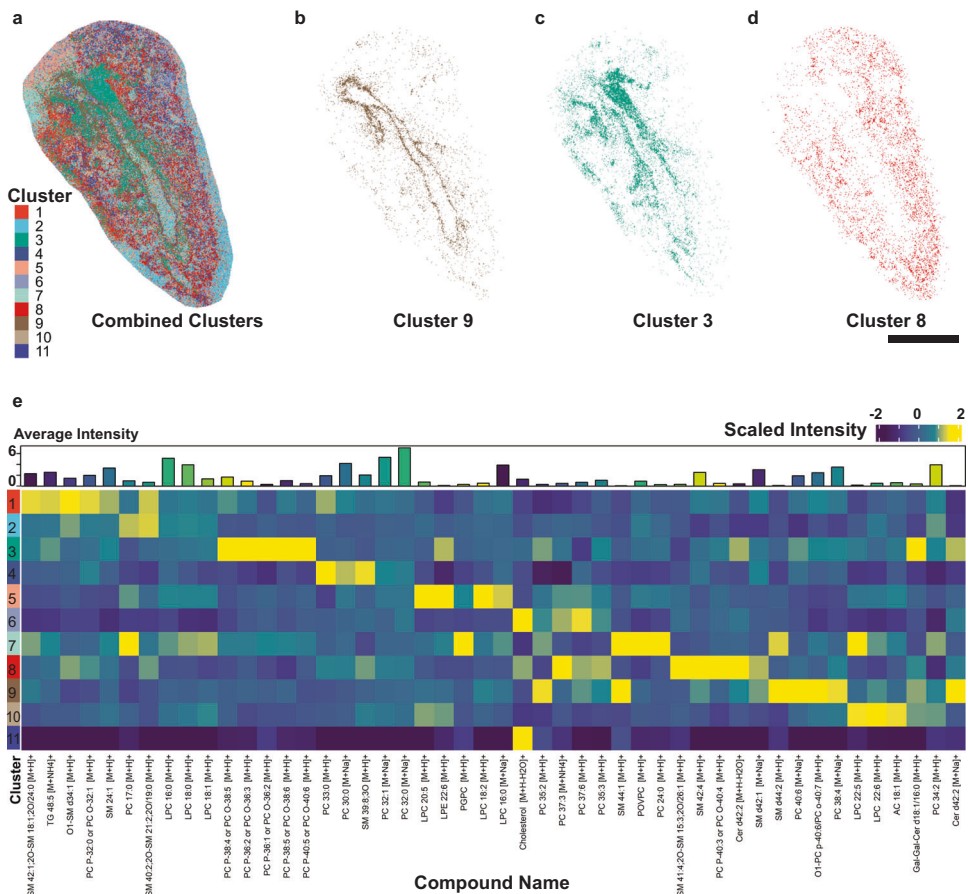

**Fig. 4 | Segmentation and colocalization analysis of annotated lipids.**
**a** Representative positive ionization mode tissue section segmented into individual regions by Seurat-based KNN clustering analysis. Extracted pixels from clusters corresponding roughly to the (**b**) airway epithelium (Cluster 9), (**c**) airway basement membrane (Cluster 3), and (**d**) alveolar epithelium (Cluster 8) from a matched H&E-stained serial section (Supplementary Fig. 2). **e** Heatmap displaying the median-scaled intensity of the top-5 lipids represented by each cluster. Lipids shared between the top-5 lists in multiple clusters are only included once. The average $\log_2$ signal intensity per pixel for each lipid is included in the top heatmap annotation. Bar = 1 mm. Source data for panel e are provided as a Source Data file.

Overall, 32 individual lipid species in females were significantly altered by HDM + $O_3$ in the alveolar epithelium, while only 13 individual lipid species were altered in the airway epithelium (Fig. 6b, d, Supplementary Data 5). Notably, we did not observe statistically significant differences in lipid class and individual species abundance between the male HDM + $O_3$-exposed and control groups in either the airway or alveolar epithelium (Supplementary Fig. 7). However, our statistical analysis was limited to a sample size of 3 per sex and treatment group

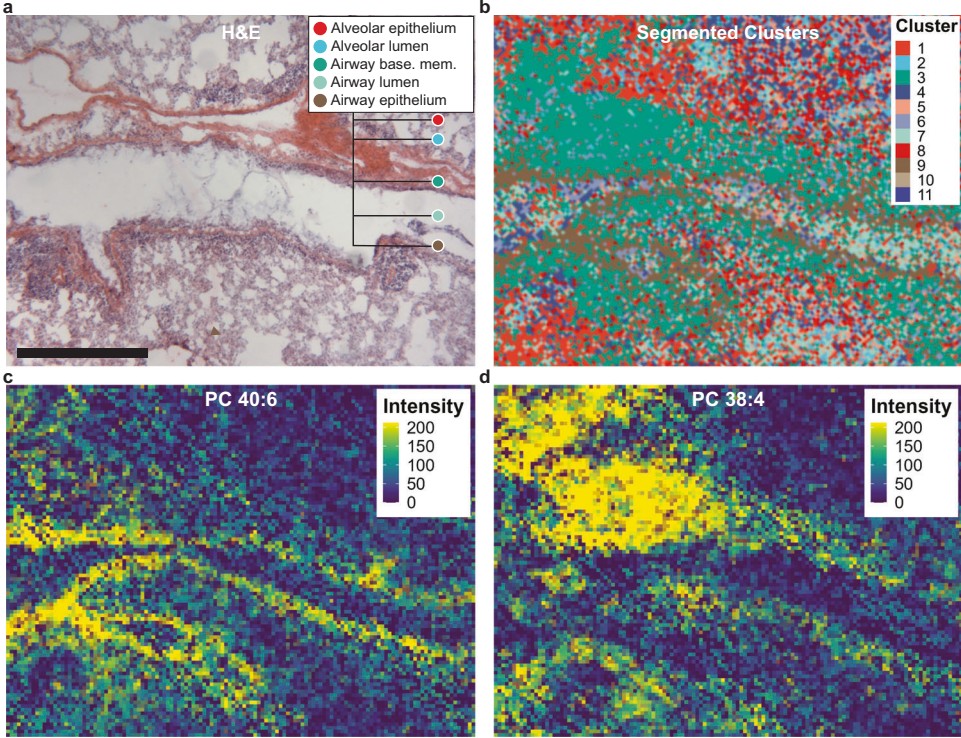

**Fig. 5 | Overlap between lung histology, unsupervised image segmentation, and individual lipid spatial distribution. a** Representative H&E-stained serial section selected from three separate sections from a male mouse exposed to HDM + $O_3$ at ×4 magnification with multiple morphological regions of interest labeled. **b** A consecutive segmented serial section acquired by MSI, including all clusters from Fig. 4. The cluster colors in (**b**) generally represent the matched regions in (**a**). **c** Ion image displaying the signal intensity and distribution of phosphatidylcholine 40:6. **d** Ion image displaying the signal intensity and distribution of PC 38:4. Both sections analyzed by MSI and H&E staining were obtained at a thickness of 15 μm. Bar = 0.5 mm.

comparing only lipids validated by LC-MS/MS, representing 10% of approximately 1800 binned peaks spanning both ionization modes (Supplementary Data 2). Notwithstanding the constraints of our analysis, these results provided proof-of-concept for multifaceted statistical comparisons across biological replicate samples in MSI data, including multiple sexes, treatment groups, and morphologically distinct regions of interest.

## Discussion

Our study extends the applications of MSI to include statistical comparisons of compound spatial distributions across biological replicates through a combination of data processing techniques and repurposed segmentation approaches. Utilization of unsupervised machine learning algorithms enabled us to characterize a model of environmental exposure at high spatial resolution within the lung. Importantly, our study provides a framework for further metabolomics studies in the lung using MSI while providing flexible analysis methods for other types of samples, including but not limited to human tissue. While previous MSI studies have recognized the need for biological replicates, many MSI studies, especially those analyzing the lung, often analyze or report data for very few tissue sections. Small sample sizes inherently preclude the ability to conduct statistical analysis and does not capture intragroup variance that may confound study results[18–20,28–30]. These limitations illustrate the challenges specific to MSI experiments related to sample preparation, data processing, and analysis throughput. Our collective methodology using an optimized agarose inflation method, sparse implementation of LOESS normalization, and unsupervised KNN clustering enabled us to elucidate the effects of multiple sexes and treatment groups across several biological replicates in specific lung regions. These results both validated and expanded upon our previous findings

analyzing microdissected airways following an identical treatment paradigm[12].

Normalization of MSI data is imperative as MSI studies progress towards increasingly complex experimental designs comparing biological replicates across multiple treatments. MALDI-MS data are susceptible to technical artifacts that skew signal intensity and overall data interpretation. Normalization methods such as TIC and vector normalization are routinely used to reduce sources of systematic drift in signal intensity such as those introduced by detector contamination across a single imaging run or by differences matrix crystal distribution[36]. Likewise, alternative methods normalizing signal intensity to an internal standard added to the MALDI matrix solution have shown promise for correcting intrasample signal variability[37]. However, additional normalization methods are needed for analyses encompassing several imaging runs that simultaneously address variability both within and across samples. Our data-driven approach of sparse LOESS normalization highlights the potential of machine learning-based methods to reduce systematic error attributed to technical artifacts in MSI data (Fig. 2, Supplementary Fig. 3). LOESS is commonly used in large-scale untargeted metabolomics assays using LC-MS/MS and is effective in mitigating systematic errors in data acquisition[38]. Applying LOESS normalization in both positive and negative mode MSI data greatly reduced variability in the TIC of each pixel within and across each sample (Fig. 2, Supplementary Fig. 3). By limiting normalization to only pixels with signal intensities greater than zero for each annotated compound, we reduced the impact of technical artifacts on subsequent image segmentation and statistical analysis while preserving biological differences in the spatial distributions of each compound (Fig. 3, Supplementary Fig. 4).

The results of our machine learning-based image segmentation reiterate the importance of sample preparation in MSI, which is critical

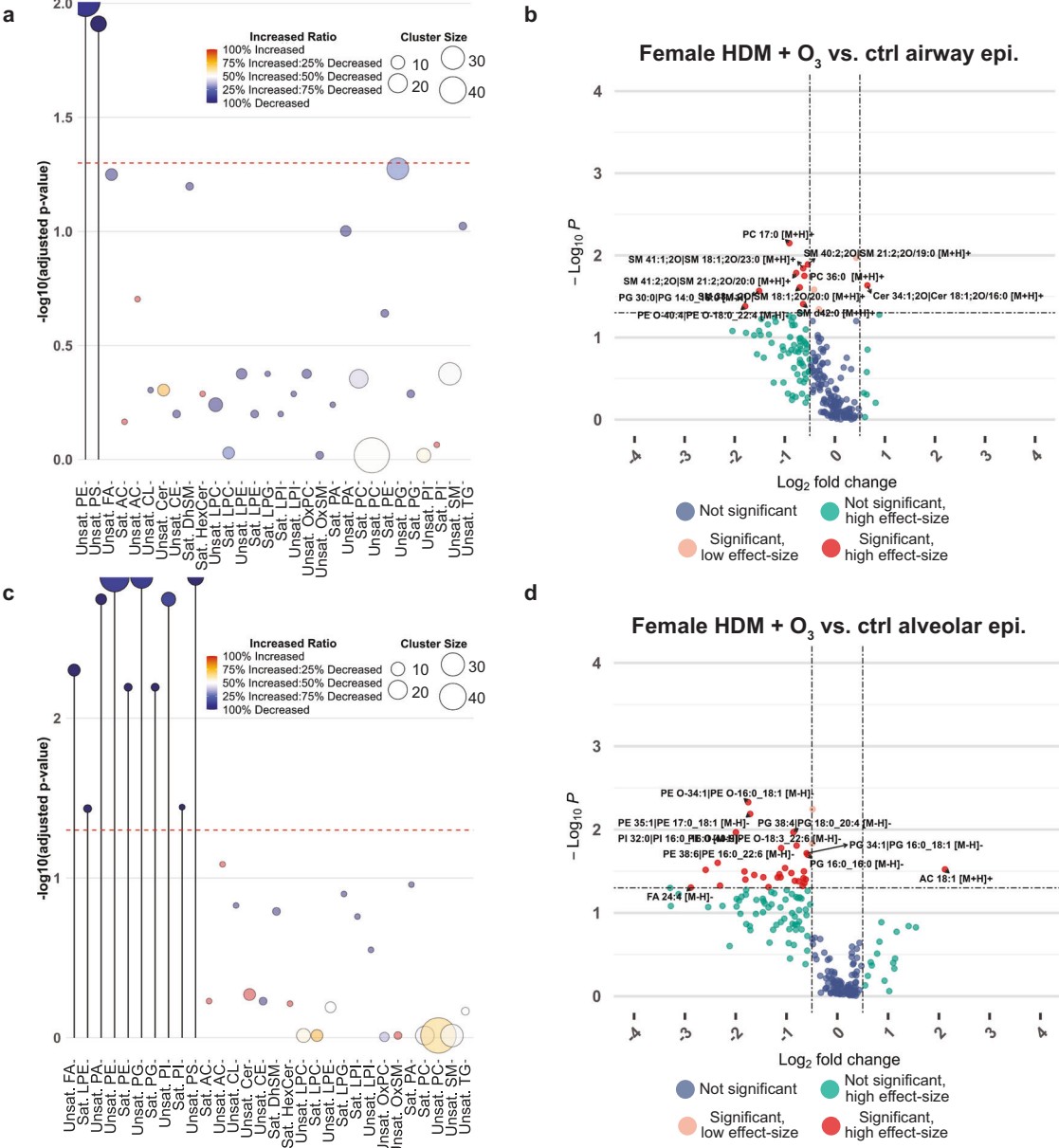

**Fig. 6 | Comparison of airway and alveolar epithelial changes in lipid composition between HDM + O₃ and control-treated female mice. a** Dot plot summarizing lipid class enrichment results comparing female HDM + O₃ and control-treated airway epithelium. Lipid classes were separated into fatty acyls with a high degree of saturation (Sat.) or a low degree of saturation (Unsat.). **b** Volcano plot summarizing significantly altered lipids in the airway epithelium comparing the HDM + O₃ group relative to female control mice. **c** Dot plot summarizing lipid class enrichment results comparing female HDM + O₃ and control-treated alveolar epithelium. **d** Volcano plot summarizing significantly altered lipids in the alveolar epithelium comparing the HDM + O₃ group relative to female control mice. All dot plots and volcano plots used a p value cutoff of p < 0.05 to determine statistical significance. The fold change direction for all panels is expressed as the abundance in the HDM + O₃ group relative to the control group. P values for enrichment

analyses were based on a one-sided Kolmogorov-Smirnov Test with FDR-correction. P values for univariate analyses were determined based on a one-way ANOVA with Tukey's post hoc analysis in R using a 95% confidence interval and default R function parameters. A log₂ fold-change of 0.5 (or a fold-change that is greater than 1) was used to define a high-effect size. Fatty acid (FA), acylcarnitine (AC), cardiolipin (CL), ceramide (Cer), cholesterol ester (CE), dihydro-sphingomyelin (DhSM), glycosylceramide (HexCer), lysophosphatidylcholine (LPC), lysophosphatidylethanolamines (LPE), lysophosphatidylglycerol (LPG), lysophosphatidylinositol (LPI), sphingomyelin (SM), phosphatidic acid (PA), phosphatidylcholine (PC), phosphatidylethanolamine (PE), phosphatidylglycerol (PG), phosphatidylinositol (PI), phosphatidylserine (PS), triacylglycerol (TG). Source data for panels a-d are provided as Source Data files.

for robust detection and spatial visualization of metabolites in relatively fragile tissue types such as the lung[39,40]. Combined agarose inflation and embedding retained cellular integrity and the capacity of our analysis to detect lipids spanning several classes across a wide m/z scan range (Fig. 1, Supplementary Fig. 1, Supplementary Data 1, Supplementary Fig. 2). We have previously validated the suitability of

agarose for mass spectrometry-based metabolomics assays, which are not compatible with other inflation media such as optimal cutting temperature (OCT) that are well known to cause ion suppression and detector contamination[11,41]. Additionally, our sample preparation method avoided washing steps used in tissue fixation or OCT removal that could potentially cause delocalization or removal of lipids while

maintaining a cryoprotective effect similar to other saccharides used in MSI studies[17,41,42]. Coupling this modified method of sample preparation with MALDI TOF-MS acquisition at a spatial resolution of 10 μm allowed us to achieve spatial information that approached the single-cell level and to segment tissue sections into distinct lung regions based on KNN clustering of annotated lipids within each sample (Fig. 4, Fig. 5, Supplementary Fig. 2)[43]. Importantly, we used the KNN algorithm implemented by the Seurat R package instead of existing segmentation methods available in the Cardinal R package or commercial software[24,34]. The classification approach used by Seurat is routinely used in scRNA-Seq studies but has not been directly applied to MSI data previously. However, we were able to reliably perform segmentation of each image through the default implementation of the Seurat clustering function after optimizing the resolution parameter. Our approach enabled segmentation of images totaling over 1.2 million pixels in negative ionization mode and 1.4 million pixels in positive ionization mode, using a combination of Seurat and RegioMSI on a computer with 128 GB of RAM in less than 5 minutes.

We validated regions segmented using Seurat by comparing our imaging data with serial sections stained with H&E, which corresponded to the epithelial and luminal compartments of both the airways and alveoli of each sample (Fig. 5, Supplementary Fig. 2). Histological confirmation of segmented images was necessary to evaluate the performance of the Seurat clustering algorithm in lieu of using a MSI reference dataset for cross-validation or marker-based approach for region assignment, both of which are currently unavailable for MSI imaging in lung tissue. Interestingly, our segmentation approach was able to further divide specific samples to isolate the airway basement membrane and the distal airway epithelium, although this degree of separation was not achieved in all study samples (Fig. 4, Fig. 5, Supplementary Fig. 8). Nonetheless, our results demonstrate the capability to compartmentalize lung MSI data into morphologically relevant regions and to visualize the abundance of individual lipids at high spatial resolution.

Our analysis of lipid spatial distribution and colocalization enabled us to confirm previous findings and to gain insights regarding lipid distribution throughout the murine respiratory tract. We observed a consistent trend across all samples irrespective of treatment or sex towards decreasing lipid saturation degree among PCs detected in positive ionization mode localized in distal compared to proximal lung regions (Supplementary Fig. 5, Supplementary Data 4). Conversely, differences in spatial distribution among lipids annotated in negative mode were largely the result of lipid class. Strikingly, polyunsaturated PEs were localized to the airway epithelium whereas saturated and unsaturated PGs were the most abundant lipid species present in the alveolar epithelium (Fig. 3, Supplementary Data 4). Differences in lipid classes of varying saturation degrees may reflect differences related to the biophysical properties of lung surfactant and mucus. PCs and PGs are abundant components of lung surfactant, which is produced by type-2 alveolar cells to reduce surface tension and to subsequently prevent alveolar collapse[44,45]. Indeed, saturated phospholipids such as dipalmitoylphosphatidylcholine (PC 32:0) are known to contribute to the biophysical characteristics of surfactant. Our MSI method detected both PC 32:0 and PA 16:0/16:0, likely an in-source fragment of PC 32:0, in the alveoli of sections analyzed in positive and negative ionization modes (Fig. 3, Supplementary Fig. 4, Supplementary Fig. 5). In contrast, an increasing degree of saturation of PEs in proximal lung regions may suggest the prioritization of surface-active properties to promote interactions between mucus and cilia[46]. Importantly, the decreased airway epithelial unsaturated PE, alveolar saturated and unsaturated PG, and saturated PE abundance resulting from HDM + O₃ exposure in females could alter baseline surfactant composition and mucus biophysical properties important for normal alveolar function and mucociliary clearance (Fig. 6, Supplementary Data 5). Further studies are needed to determine the

functional relationship between lipid saturation and biophysical properties throughout the respiratory tract under normal conditions and under toxicant exposure.

The spatial distribution of individual lipid species may further distinguish the functional differences between specific lung regions. SM 44:1, SM 44:2, and docosahexaenoic acid were all highly localized to the airway epithelium across all samples, the abundances of which were independent of treatment or sex (Fig. 4, Supplementary Data 4). Sphingolipids containing very long chain fatty acids (VLCFA) have previously been implicated in modifying immune responses involving macrophages and natural killer T (NKT) cells[47,48]. Furthermore, docosahexaenoic acid administration in vivo has previously demonstrated a protective role in a bleomycin-induced model of pulmonary fibrosis in part from reduced cellular inflammation within the lung[49]. Therefore, the distribution of VLCFA SM lipids and docosahexaenoic acid in the airway epithelium may serve to modulate the activity and resolution of local immune responses in the airways. Our findings could be further explored by characterizing individual immune cell types in tandem with targeted lipidomics assays to confirm if the individual SM lipid species we identified are associated with these responses.

The results of our lipid class-based enrichment analysis revealed decreases in unsaturated PS abundance in both the airway and alveolar epithelium of female mice treated with HDM + O₃ (Fig. 6, Supplementary Data 5). Decreased phosphatidylserines within both regions could inhibit signaling events related to macrophage function that would favor the resolution of severe inflammation present in our combined exposure model. Phosphatidylserines are immunosuppressive mediators that are expressed on apoptotic cells and recognized by phagocytes during efferocytosis[50–52]. While the role of phosphatidylserines is largely studied within the context of viruses and cancer cells leveraging phosphatidylserine signaling in immune cell evasion, the T cell/transmembrane, immunoglobulin, and mucin (TIM) family of proteins respond to phosphatidylserine and have been investigated for their roles in modulating T-cell responses in asthma[53]. Thus, HDM + O₃-associated decreases in phosphatidylserine abundance we observed may indicate a mechanism promoting inflammatory signals resulting in severe inflammation and immune cell influx modeled in our previous study (Fig. 6, Supplementary Data 5)[12].

Lastly, we observed significant differences in airway epithelial sphingolipid abundance in female mice. Specifically, 6 long chain sphingomyelins containing 2 or fewer double bonds were significantly decreased in HDM + O₃ exposed airway epithelium, which was accompanied by an increase in ceramide 34:1 (Fig. 6, Supplementary Data 5). This result confirms our previous finding from microdissected airway tissue that decreases in sphingomyelins are concomitant with increased ceramide following combined HDM + O₃ exposure relative to saline and filtered air controls[12]. Importantly, these changes induced by combined HDM + O₃ exposure may indicate an increased breakdown of sphingomyelin via acid sphingomyelinase, which cleaves cell membrane-bound sphingomyelin into ceramide that can undergo further breakdown into sphingosine 1-phosphate or modification into glycosphingolipids[54]. Sphingolipid signaling through ceramide and sphingosine 1-phosphate is well characterized to modulate several hallmark features of allergic asthma, including airway hyperresponsiveness, airway inflammation, and immune cell influx to the airways[55,56]. Unfortunately, our MSI method was not optimized to detect lower m/z compounds including sphingosine 1-phosphate and only detected one glycosphingolipid validated with LC-MS/MS, so we were unable to comprehensively evaluate possible changes in sphingosine 1-phosphate and glycosphingolipid abundance following combined HDM + O₃ exposure (Supplementary Data 2). Nonetheless, the increased spatial resolution afforded by our MSI analysis enabled us to identify specific locations within the lung targeted by HDM + O₃ compared to microdissected tissues.

The limitations of our study warrant future experiments incorporating additional biologic replicates, improved annotation of detected peaks, and orthogonal approaches to provide a holistic understanding of the importance of the spatial lipidomic changes we observed in ozone-mediated exacerbation of allergic asthma. Greater variance in lipid abundance within treatment groups and coverage of the lipidome by our MSI assay likely contributed to the absence of statistically significant alterations in lipids observed in males compared to females (Fig. 1, Supplementary Data 1, Supplementary Data 2). While our analysis methods addressed throughput limitations presented by spatial data processing, segmentation, and statistics, optimization of MSI sample preparation is needed to further increase the sample size and statistical power of MSI studies. Consequently, we were unable to perform MSI on lung sections collected from mice treated with HDM or ozone alone to assess additive or synergistic responses in our study. We previously determined that combined HDM + O$_3$ exposure induced comprehensive changes in the lipidome of microdissected airways, increased airway hyperresponsiveness, and airway inflammation relative to control-treated mice[12]. Conversely, enrichment analyses evaluating changes in lipid class abundance in microdissected airways treated with HDM or exposed to ozone were modest relative to our previous results (Supplementary Fig. 9). Considering these findings, we primarily focused on evaluating spatial changes in lipid abundance in the combined HDM + O$_3$ group.

Likewise, the technical aspects of our MSI data acquisition could be further optimized to improve quantitative image analyses. Alternative normalization methods combining the strengths of our sparse LOESS normalization with using series of internal standards added to each sample may further attenuate technical variance in our data. Additionally, our statistical analyses of ion images were based solely upon lipids validated by LC-MS/MS. While this method of compound annotation is more reliable than assigning peak identities based on m/z alone, roughly 10% of the binned peaks detected across both ionization modes were annotated in total. Using tandem mass spectrometry (MS/MS) and trapped ion mobility spectrometry are needed to further increase lipidome coverage to complement lipid annotations based on LC-MS/MS validation. Such approaches could be used to distinguish more detailed peak information, including the position of double bonds on lipid acyl chains. Existing instrument software is not currently capable of simultaneous acquisition of MS and MS/MS spectra, which means that characteristic fragmentation information that assists with identification must be collected on a sample after acquisition for each feature individually. Therefore, software improvements enabling simultaneous MS/MS acquisition are critical for metabolomics assays based on MSI considering the hundreds of individual features we detected in our study. However, our use of sparse LOESS normalization and extensive filtering of detected MSI peaks both outperformed conventional TIC normalization of our data and limited analyses to lipids annotated with high confidence.

Finally, we were unable to isolate all morphologically defined lung regions for all samples using our unsupervised segmentation approach, including the airway basement membrane and the distal airways (Figs. 3, 4, 5, Supplementary Fig. 4). Increasing the number of samples in our study and varying the sectioned location of tissue blocks may have increased the number of samples containing clearly defined basement membranes and distal airway generations. Characterization of these regions across treatments is needed to determine localization and changes in lipids within the specialized cell types located within each region and how these changes contribute to lung disease[9]. Nonetheless, our segmentation results facilitated unbiased regional comparisons in lipid abundance that highlight the potential of MSI to study molecular differences within the lung and how these regions are impacted by inhaled toxicants.

We have characterized a model of environmental exposure in the lung using mass spectrometry imaging. This technique revealed striking differences in lipid distribution between regions of lung tissue sections at high spatial resolution. The modified agarose-inflation for lung cryosectioning coupled with the comprehensive data processing, image segmentation and analysis strategies included the RegioMSI package presented here provide the foundation for future MSI experiments not only in the lung but also for other MSI studies. Our approach could be applied to investigations in human tissue or additional mechanistic rodent studies examining regional effects of other air pollutants within the lung such as traffic-related air pollution and wildfire smoke. Additionally, the detailed spatial information we acquired from this study may implicate previously overlooked cell types and local lipid composition changes in signaling pathways that could subsequently be targets for therapeutic development in severe asthma. Importantly, our lipidomic analysis using MSI could also be paired with scRNA-Seq and spatial transcriptomics techniques to confine the changes we observed in lung lipids to individual cell types and evaluate cell-type specific gene expression to enhance our understanding of the molecular determinants of lung disease.

## Methods

### Ethical statement
All animal exposures and experiments were conducted following approved protocols (protocol number 22219) reviewed by the UC Davis Institutional Animal Care and Use Committee in accordance with guidelines for animal research established by the National Institutes of Health.

### Animal protocol
Adult male and female BALB/c mice (Envigo, Inc.) 8–10 weeks of age were acclimatized for 1 week in rooms kept on a 12 h/12 h light/dark cycle and fed Purina 5001 lab diet. Mice were sensitized to crushed whole bodies of *Dermatophagoides farinae* (Stallergenes Greer, Inc.) dissolved in phosphate-buffered saline (PBS), challenged with HDM, and acutely exposed to ozone (Fig. 1a). The sensitization phase consisted of three intranasal instillations of 10 µg HDM in 25 µL PBS or vehicle on Days 1, 3, and 5 followed by three consecutive challenges with HDM (10 µg in 25 µL PBS) on Days 12–14[57]. Following each HDM challenge on Days 12–14, mice were exposed to either filtered air (FA) or ozone (0.5 ppm, 6 hr/day). Exposure chambers were individually monitored every hour by a Teledyne Model 400E ozone analyzer to ensure stability of ozone concentrations for the entire duration of exposure. The left lung lobes from each mouse were cannulated and inflated 24 h after the final ozone exposure with 1% low-melting temperature agarose to isolate microdissected airways (n = 8 M & 8 F/group) or to prepare lung lobes for MSI, which were placed on ice for 10 min in PBS to solidify (Supplementary Fig. 1)[10]. Each lobe designated for MSI was cut into two transverse sections and embedded in 1% low-melting temperature agarose, and the tissue blocks were sealed and submerged in liquid nitrogen-cooled isopentane (n = 3 M & 3 F/group, total n = 12). Embedded tissue blocks or microdissected airways were stored promptly at -80 °C until cryosectioning or extraction.

### Lung histology
Left lung lobes were fixed with 1% paraformaldehyde in PBS for 10 min. The fixed serial sections were dehydrated, stained with H&E, and subsequently imaged using an Olympus BH-2 microscope with a 4x and 10x objective. Representative images were selected to compare sections analyzed by MSI and to identify morphologic changes in response to combined HDM + ozone exposure (n = 3 males and 3 females per study group; total n = 12).

### Cryosectioning
Each tissue block was transferred to a Leica CM1950 cryostat and mounted to a metal chuck using optimal cutting temperature (OCT) medium (Sakura Finetek USA). The OCT was only applied to one face of

the tissue block to prevent detector contamination. Sections 15 µm thick were cut and thaw-mounted onto indium tin oxide (ITO) coated glass slides (Delta Technologies, Limited), and serial sections were thaw-mounted onto poly-L-lysine coated glass slides for hematoxylin and eosin (H&E) staining. Tissue sectioning was also tested at 5 and 10 µm but this thickness was determined unsuitable for downstream MSI data acquisition due to the presence of folding and scoring artifacts present in tissue sections. The ITO slides were then promptly stored at −80 °C until matrix application.

### Matrix application and MSI data acquisition

Two sets of ITO slides were prepared for either positive or negative ionization mode data acquisition. Slides designated for positive mode ionization were sprayed with a solution containing 40 mg/mL dihydroxybenzoic acid (Millipore Sigma) in 70:30 MeOH:$H_2O$ using an HTX automatic sprayer system. The parameters for the sprayer were set to 75°C nozzle temperature, 8 passes, 0.1 mL/min. flow rate, 2 mm track distance, and 10 second drying time between each pass. Slides acquired by negative mode ionization were sprayed with 7 mg/mL 1,5-diaminonaphthalene dissolved in 70:30 MeOH:$H_2O$. Both matrices and application parameters were optimized for broad lipidome coverage in each ionization mode without targeting a specific lipid class[39]. The sprayer parameters were set to 70 °C nozzle temperature, 13 passes, 0.1 mL/min. flow rate, 2 mm track distance, and 10 second drying time between each pass. Matrix-coated slides were analyzed in positive and negative ionization mode by matrix-assisted laser desorption/ionization time-of-flight mass spectrometry (MALDI TOF MS) using a Bruker timsTOF fleX hybrid trapped ion mobility time-of-flight mass spectrometer equipped with a MALDI source. Ion mobility separation was disabled for all experimental runs and samples were randomized for acquisition in both ionization modes. Prior to analysis, the detector was calibrated using a solution of 90% Agilent ESI-TOF tuning mix and 10% sodium formate. Automatic target profile and laser focus adjustments were performed upon loading slides, and the laser raster width was set to 10 µm to acquire data using 1 burst of 150 shots at a frequency of 10,000 Hz. Beam scan was disabled and a laser field size of 5×5 µm was used to acquire data at 50% local laser energy and 0% global attenuator offset. The detector scan range for positive ionization mode was 300-1300 m/z and 300-900 m/z for negative ionization mode.

### Sample preparation and LC-MS/MS data acquisition

Following MALDI-TOF MS data acquisition, matrix-coated tissue slides were scraped, and lipids were extracted for untargeted lipidomics analysis by LC-MS/MS. Microdissected airways were acquired separately but followed the same extraction protocol as scraped tissue samples. Specifically, a biphasic extraction mixture consisting of 225 µL methanol and 750 µL methyl *tert*-butyl ether containing 76 internal lipid standards (Avanti Polar Lipids UltimateSPLASH ONE kit plus acylcarnitines and free fatty acids) was used to extract lipids from scraped tissues[11,12,58,59]. Briefly, the top fraction was evaporated to dryness and resuspended in 90 µL of 90:10 MeOH:Toluene with 50 ng/mL 12-[(Cyclohexylcarbamoyl)amino]dodecanoic acid (CUDA), and analyzed by a ThermoFisher Scientific Vanquish UHPLC+ liquid chromatography system coupled to a Q-Exactive HF orbital ion trap mass spectrometer in both positive and negative ionization modes. One experimental sample from each condition ($n = 1$/group/sex, total $n = 4$) was acquired in each mode along with two extraction blanks and two technical replicates of pooled experimental samples. A Waters Acquity UPLC CSH C18 column and a mobile phase consisting of 60/40 v/v acetonitrile:water (A) and 90/10 v/v isopropanol:acetonitrile (B) were used to separate metabolites for lipidomics analysis. Formic acid (0.1%) and ammonium formate (10 mM) were used as modifiers for positive mode acquisition, and ammonium acetate (10 mM) was used as a modifier for negative mode. The data acquisition parameters

for both positive and negative ionization modes were: 65 °C column chamber temperature, 65 °C post-column cooler temperature, 65 °C column preheater temperature, 5-minute acquisition time, and stepped normalized collision energies of 20, 30, and 40%. The acquisition mass ranges were 120-1700 m/z and 113.4-1700 m/z for positive and negative ionization modes, respectively.

### LC-MS/MS data processing

Processing of raw LC-MS/MS files, including deconvolution, peak picking, and alignment, was completed in MS-DIAL v.4.70[60]. Identification for all compounds was based on mass spectra from built-in in silico libraries[61]. Matches to in silico libraries were based on m/z, retention time, and MS/MS fragmentation pattern. The curated annotation list for positive and negative ionization mode is available in Supplementary Data 1.

### MSI data processing, normalization, and statistical analysis

Raw data files were imported into SCiLS Lab software (Bruker Daltonics, Inc.) with a 5-ppm bin size for exporting in imzML format. The preprocessed imzML files containing all raw data were subsequently imported into R Studio (v.2023.06.2 + 561) or directly into R (v.4.3.3, v.4.4.1, and v.4.4.2) for further processing and downstream analysis. Processing steps including peak detection, binning, and alignment were completed using the R package Cardinal (v3.4.3 or v3.6.2)[31]. MSI peak annotations were assigned based on matching m/z values to untargeted LC-MS/MS-based lipidomics data from scraped tissue slides and data from microdissected lung tissue under identical experimental conditions as previously reported[12]. Images corresponding to technical artifacts from the assigned annotation list were then removed. The individual annotation list, including the mass error between the MSI peak list and LC-MS/MS validated annotations, is included in Supplementary Data 1. Additionally, a summary of annotations stratified by class, subclass, and saturation from positive and negative mode is included in Fig. 1b and Supplementary Data 2.

MSI data were normalized by adapting locally estimated scatterplot smoothing (LOESS), a local polynomial regression algorithm, to all non-zero intensity pixels across all samples. Image segmentation and clustering were completed by implementing the k-nearest neighbors (KNN) algorithm and graph-based clustering approach used by the R package Seurat (v5.0.3 and v5.1.0)[34]. All default function parameters were used for clustering except the clustering resolution, which was set to 0.9 for all samples. The segmentation results were then assigned to morphological regions of interest by comparison to H&E-stained serial sections of each tissue block. Lastly, univariate and multivariate statistical analysis were conducted using various R packages and the Kolmogorov-Smirnov test to conduct enrichment analysis for each lipid class as previously described[35,62]. Specifically, individual lipids were grouped based on similarities in chemical structure to determine if a specific class of lipids was significantly enriched between two groups. All analysis scripts are freely available online as a separate R-package in GitHub [https://github.com/cschasestevens/RegioMSI] or compiled in the Zenodo database under the accession code 10.5281/zenodo.14834145 [https://zenodo.org/records/14834145]. RegioMSI was tested on Windows 10 and Windows subsystem for Linux (WSL2) using a Windows PC equipped with a 12-Core 4.10 GHz processor and 128 GB of RAM.

### Reporting summary

Further information on research design is available in the Nature Portfolio Reporting Summary linked to this article.

## Data availability

The raw and processed data generated in this study have been deposited in the Zenodo database under the accession code

10.5281/zenodo.14846221 [https://zenodo.org/records/14846221]. The data are available free of restrictions and can be downloaded directly from Zenodo. Source data for all graphs included as figure panels within this study have been deposited in the Zenodo database under the accession code 10.5281/zenodo.14846581 [https://zenodo.org/records/14846581]. Supplementary tables are included as Supplementary Data files with this manuscript. Unless otherwise stated, all data supporting the results of this study can be found in the article, supplementary, and source data files.

## Code availability

All source code used for the analysis and visualization of the study data are available on Github [https://github.com/cschasestevens/RegioMSI] under a MIT license and can be compiled as a R package for local use following the instructions outlined in the RegioMSI package documentation[63]. Package dependencies are freely available from either the Comprehensive R Archive Network [CRAN: https://cran.r-project.org/] or Bioconductor [https://www.bioconductor.org/]. A compiled version of the RegioMSI source code is deposited in the Zenodo database under the accession code 10.5281/zenodo.14834145 [https://zenodo.org/records/14834145].

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

## Acknowledgements

L.S.V.W. is supported by National Institutes of Health (R21 ES030276 and P30 ES023513). N.C.S. and V.J.B.E. received support from T32 ES007059 and M.C.D., N.C. S. and V.J.B.E. were supported by T32 HL007013. Lipid identifications were supported by U2C ES030158 to Oliver Fiehn. The timsTOF mass spectrometer was purchased through S10 OD030253 to O.F. We thank the undergraduate student members of the Van Winkle laboratory for their assistance with lung microdissection and support during sample collection. We thank Brandon Tran, an undergraduate student in the Van Winkle laboratory, for his contributions to the staining of the H&E slides prepared for this study.

## Author contributions

N.C.S., L.S.V., and O.F. prepared the manuscript; N.C.S. and L.S.V. developed the experimental design for the study; N.C.S., V.J.B.E., M.C.D., and J.M. performed the animal experiments and tissue collection; T.S. acquired LC-MS/MS data for validation and assignment of imaging data annotations; N.C.S. conducted imaging data acquisition, processing, statistics, and analysis; E.K.N. provided training and technical expertise in sample preparation and data acquisition; and all authors contributed to editing the manuscript.

## Competing interests

The authors declare no competing interests.
