## [Transparent Peer Review file · Nature Communications]

Resolving multi-image spatial lipidomic responses to inhaled toxicants by machine learning

Corresponding Author: Professor Oliver Fiehn

Version 0:

Reviewer comments:

Reviewer #1

(Remarks to the Author)

The investigators have used high resolution mass spectrometry imaging to examine spatial differences in lipid distribution in the mouse lung following HDM sensitisation and ozone exposure. Using what looks like having been done for the first time, the images (validated by LC-MS/MS data from tissue) were segmented, allowing for comparison of specific lung regions such as airways versus alveoli. There are significant differences in sphingolipid and glycerophospholipid composition shown, particularly after hdm and ozone in female mice (but not in males). This represents an important progress in quantification at lung tissue level of lipid composition following various insults to the lungs.

My major comment here is related to the comment that “relatively modest effects of either HDM or ozone alone is significantly altering global lipid abundance” precluded the investigators from studying the mice exposed only to HDM or ozone at all. Despite the fact the investigators have shown significant and important differences in the airways and alveolar regions in terms of lipidomics comparing HDM and ozone mice to healthy mice, they should compare HDM and ozone mice to the hdm only and the ozone only mice, to confirm similar (or may be different) differences. This would be a post-hoc analysis and it would be sufficient to do a direct statistical comparison rather than needing a multiple comparison approach that would not be possible with only $n = 3$ in each group.

My other comment is related to a discussion of why the phospholipid comparison alteration differs between airways and alveoli. This is important given that the investigators have already a good discussion of the significance of their findings. The strength and limitations of their novel technique are also well covered.

(Remarks on code availability)

I have not seen any code attached.

Reviewer #2

(Remarks to the Author)

Remarks to the Author

In this manuscript, the authors evaluate lipid distribution in mouse lung subject to combined treatment of allergen sensitization and ozone exposure. They identify spatially distinct lipids with varied degrees of saturation between distal and proximal regions. They also identified abundances of sphingolipid and glycerophospholipid between the epithelium of the airway and alveolar, while the difference only exists in females. This study demonstrates a novel method for imaging spatial responses to inhaled toxicants in lung, employing modified agarose-inflation for lung cryo-sectioning coupled with comprehensive data processing, image segmentation and analysis strategies. However, the contents are not well organized, and the study exhibits a lack of biological significance. Thus it is unlikely to get interest of broader readership. Several issues should be addressed.

1. Contents in “Software and code” of the Reporting Summary file are not relevant to the current manuscript.
2. For better understanding, I recommend that the physiological and pathological changes elicited by inhaled toxicant should be provided in Introduction. Also the model of HDM/ozone should be introduced. The information on whether the HDM/ozone is representative for pathological changes of “inhaled toxicants” as stated in the title should be described, and the reason for selecting this model instead of other toxicants should be provided.
3. Relevant literatures on application of the KNN algorithm or machine learning in image segmentation should be briefly

reviewed. MSI preprocessing methods in addition to sparse LOESS normalization should be reviewed.

4. As mentioned in the Introduction section, responses to inhaled toxicants in lung may confine to specific cell type of populations. How is the cell type annotated in MSI data? Are there cell type markers for assignment of each pixel similar to that in scRNA-seq analysis to label each cell type, such as epithelial cells and immune cells?
5. What is the major change in individual cell types when exposed to inhaled toxicant? Are the spatially distinct lipids related to these cell types? E.g. are polyunsaturated PEs spatially correlated with eosinophils, neutrophils or related cytokines? These should be determined in serial sections.
6. As stated in line 301, "displayed distributions that appeared to be independent of potential effects induced by treatment or sex", In figure S4 corresponds to Figure 3, all samples are shown, whereas large variance seemed to exist among individual samples. Detailed information of individual samples including sex and ozone exposure should be provided. And the impact of ozone exposure on these three lipids should be discussed.
7. In Figure 3, are PE 18:0/22:6, PI 36:4, and PA 16:0/16:0 the top 3 distinct lipids? and how are they selected from all annotated lipids as examples? Why is not PI 36:4 annotated at the acyl chain level as the other two?
8. Which cluster in Figure 4A(cluster1-11) do pixels in Figure 4B-D correspond to? the information should be labeled on figures or in legends.
9. Comparing Fig. 5B-D to 5A, it seems that MS signals exist in the airway lumen, otherwise the signal may be attributed to a much thinner lumen cavity in the sections of Fig. 5B-D compared to 5A. The authors should state whether the sections are consecutive and explain the reason underlying the variation.

(Remarks on code availability)

Reviewer #3

(Remarks to the Author)

In this manuscript, the authors provided a framework for future mass spectrometry imaging experiments capable of relative quantification across biological replicates and expansion to multiple sample types. It was a great exploration and worth publishing in Nature Communications. Before the manuscript is published, I still have the following concerns.

1. As in your manuscript, following MALDI-TOF MS data acquisition, matrix-coated tissue slides were scraped, and lipids were extracted for lipidomic. Do you consider the effect of MALDI matrix for MS detection?
2. The choice of MALDI matrix is a critical factor for lipid detection. Did you evaluate different matrices for lipid detection in your study?
3. You applied sparse LOESS normalization to minimize systematic errors while preserving biological differences in the spatial distributions of each compound. LOESS normalization is commonly used in LC-MS/MS-based metabolomics assays. Could you elaborate on why LOESS normalization is advantageous in MSI data, especially when addressing variability within and across samples? Additionally, how does LOESS compare to other common normalization methods in terms of its specific strengths and limitations for MSI?
4. You mentioned that the use of agarose inflation and embedding preserved the cellular integrity of lung tissue and avoided ion suppression and detector contamination caused by OCT. However, the degree of separation between the airway basement membrane and distal airway epithelium varied among samples. Are there any additional optimization steps that could be taken to improve the consistency of this separation, especially when applying the method to other more complex or fragile tissue types?
5. Can mass spectrometry imaging distinguish the specific locations of double bonds in lipids? Did you consider double bond positions in the development of this method?
6. Does mass spectroscopic imaging require more regional selection of tissue samples? In this experiment, how to ensure the consistency of the selection of lung tissue regions?
7. In this experimental model, sphingolipid and glycerophospholipid abundance showed differences in male and female mice. Have there been relevant reports before? Do your results have implications for the construction of such lung disease models in the future?

(Remarks on code availability)

Version 1:

Reviewer comments:

Reviewer #1

(Remarks to the Author)

The authors have answered the concerns I raised to my satisfaction.

(Remarks on code availability)

the code is accessible and runs.

Reviewer #2

(Remarks to the Author)

The revised manuscript provides readers with a more comprehensive research context and a more in-depth discussion of significance and limitations. Major concerns have been addressed.

Minor Points

1. In the last paragraph of the Methods section, on line 234, "as previously" should be changed to "as previously described."
2. The labeling in Figure 6 should be enlarged and simplified for better visualization. For better understanding, the lipid classes should be labeled in the same form as mentioned in the main text, e.g., PE, PS, with abbreviations included in the legends.

(Remarks on code availability)

The attached code appears to be of sufficient quality and accessibility, and it functions correctly with the demo data for two samples. However, since the demonstration of the RegioMSI package includes only a subset of the data, as noted in the Reporting Summary file, it is not possible to fully evaluate whether the results presented in the manuscript are reproducible. In addition, the object 'd' in the peak processing code is not clearly defined or explained, which may cause confusion.

Reviewer #3

(Remarks to the Author)

I think the authors have answered my concerns well enough to publish in nature communications.

(Remarks on code availability)

We thank the reviewers for their helpful comments and suggested edits to our manuscript. We have revised the manuscript accordingly and have added additional figures, text, and references as requested.

Reviewer #1 (Remarks to the Author):

The investigators have used high resolution mass spectrometry imaging to examine spatial differences in lipid distribution in the mouse lung following HDM sensitisation and ozone exposure. Using what looks like having been done for the first time, the images (validated by LC-MS/MS data from tissue) were segmented, allowing for comparison of specific lung regions such as airways versus alveoli. There are significant differences in sphingolipid and glycerophospholipid composition shown, particularly after hdm and ozone in female mice (but not in males). This represents an important progress in quantification at lung tissue level of lipid composition following various insults to the lungs.

Rev1, Comment 1: My major comment here is related to the comment that “relatively modest effects of either HDM or ozone alone is significantly altering global lipid abundance” precluded the investigators from studying the mice exposed only to HDM or ozone at all. Despite the fact the investigators have shown significant and important differences in the airways and alveolar regions in terms of lipidomics comparing HDM and ozone mice to healthy mice, they should compare HDM and ozone mice to the hdm only and the ozone only mice, to confirm similar (or may be different) differences. This would be a post-hoc analysis and it would be sufficient to do a direct statistical comparison rather than needing a multiple comparison approach that would not be possible with only $n = 3$ in each group.

Response to Rev1, Comment 1: We have now included new data on exposures with either HDM or O₃ alone. Results are presented as lipid class-based enrichment analyses comparing the HDM-treated or ozone-exposed mice to control-treated microdissected mouse airways as **Figure S9**. Similar to our previous findings (Stevens et. al. 2023), we observed fewer lipid classes that were significantly increased or decreased in abundance when exposed to either HDM or O₃ alone. We would like to note that exposure doses that we present here more closely mimic a realistic model of exposure, in contrast to higher doses that were used by other authors (see doi: 10.1016/j.biopha.2018.02.079, 10.1016/j.ejphar.2008.09.031). Overall, however, we still focus on combined exposure of HDM and O₃ which are known as the classic model of ozone-induced exacerbation of asthma (added now in line 58ff), as published multiple times and cited in our manuscript. As noted by the reviewer, we have also now added the limited possibilities of performing advanced statistics analyses due to the low number of independent samples per group .

Rev1, Comment 2: My other comment is related to a discussion of why the phospholipid comparison alteration differs between airways and alveoli. This is important given that the investigators have already a good discussion of the significance of their findings. The strength and limitations of their novel technique are also well covered.

Response to Rev1, Comment 2: Increasing lipid saturation in distal lung regions, including the terminal bronchioles and alveoli, may correspond to lower surface tension, thereby preventing collapse during respiration. In contrast, cartilaginous airways in the proximal regions of the lung would not be susceptible to collapse, suggesting that lipids in these regions may modulate different biophysical properties than in the distal lung. We have now included an additional sentence in the main text highlighting the need for additional studies to determine the functional relationship between regional lung lipid composition and biophysical properties. We are currently engaged in these studies using an *in vitro* culture model that will measure regional lipid composition and biophysical properties of apical mucus and surfactant.

Rev1, Comment 3: I have not seen any code attached.

Response to Rev1, Comment 3: We have updated the Reporting Summary file as well as the Software and Code Submission checklist with relevant information pertaining to the code used in our manuscript. All

source code can be compiled and installed as a R package from the URL provided in the main text. Specific installation instructions and hardware requirements are included at this URL for reference.

Reviewer #2 (Remarks to the Author):

In this manuscript, the authors evaluate lipid distribution in mouse lung subject to combined treatment of allergen sensitization and ozone exposure. They identify spatially distinct lipids with varied degrees of saturation between distal and proximal regions. They also identified abundances of sphingolipid and glycerophospholipid between the epithelium of the airway and alveolar, while the difference only exists in females. This study demonstrates a novel method for imaging spatial responses to inhaled toxicants in lung, employing modified agarose-inflation for lung cryo-sectioning coupled with comprehensive data processing, image segmentation and analysis strategies. However, the contents are not well organized, and the study exhibits a lack of biological significance. Thus it is unlikely to get interest of broader readership. Several issues should be addressed.

Rev2, Comment 1: Contents in “Software and code” of the Reporting Summary file are not relevant to the current manuscript.

Response to Rev2, Comment 1: We have now updated the Reporting Summary file with the correct information pertaining to this manuscript and have provided further documentation as a README at the provided URL where the package may be installed.

Rev2, Comment 2: For better understanding, I recommend that the physiological and pathological changes elicited by inhaled toxicant should be provided in Introduction. Also the model of HDM/ozone should be introduced. The information on whether the HDM/ozone is representative for pathological changes of “inhaled toxicants” as stated in the title should be described, and the reason for selecting this model instead of other toxicants should be provided.

Response to Rev2, Comment 2: The introduction has been updated with a brief summary of the acute effects of ozone exposure and its relevance as a region-specific toxicant. We now also elaborate on the combined HDM/ozone exposure model and the rationale for using this model as it is well characterized and relates to our previous findings.

Rev2, Comment 3: Relevant literatures on application of the KNN algorithm or machine learning in image segmentation should be briefly reviewed. MSI preprocessing methods in addition to sparse LOESS normalization should be reviewed.

Response to Rev2, Comment 3: We have now highlighted references within the main text that explain segmentation methods available from the Cardinal R package. The Seurat implementation of KNN has not been applied to MSI data previously. However, using our RegioMSI package concurrently with Seurat reliably segmented all images across ionization modes in our MSI dataset, which included over 1.2 million pixels in negative ionization mode and 1.4 million pixels in positive ionization mode. Both Seurat and Cardinal use unsupervised classification algorithms to perform clustering. In our experience, however, we were unable to use Cardinal for classification due to the computational resources required by the package whereas we did not encounter resource limitations using Seurat. We have now included additional information supporting our choice to use Seurat in the discussion section and have added additional context regarding LOESS normalization in the method section, which has been routinely used in LC-MS/MS datasets and is a method of local polynomial regression. **Figure 2** and **Figure S3** compare the performance of TIC normalization, which is a common method of MSI normalization, and sparse LOESS (sLOESS).

Rev2, Comment 4: As mentioned in the Introduction section, responses to inhaled toxicants in lung may confine to specific cell type of populations. How is the cell type annotated in MSI data? Are there cell type

markers for assignment of each pixel similar to that in scRNA-seq analysis to label each cell type, such as epithelial cells and immune cells?

Response to Rev2, Comment 4: To our knowledge, there are no studies of lung lipids at true single-cell resolution with MS-imaging, along with true single-cell analysis of cell types. Here, single-cell RNAseq would be the best option to use for cell-type analyses. No such study is known to the authors for lungs. Therefore, we did not annotate cell types. Instead of assigning individual cell types, we here annotated regions of interest by functional units such as alveolar epithelium, or airway epithelium, using the H&E images with respect to histological validations. Likely, even higher resolution than 10 μm that was achievable in our work would not have been sufficient to clearly delineate individual cell types.

Rev2, Comment 5: What is the major change in individual cell types when exposed to inhaled toxicant? Are the spatially distinct lipids related to these cell types? E.g. are polyunsaturated PEs spatially correlated with eosinophils, neutrophils or related cytokines? These should be determined in serial sections.

Response to Rev2, Comment 5: As mentioned previously, the limitations of our instrumentation and the lack of accepted marker lipids prevented us from identifying individual cell types in our study. Additionally, the agarose inflation step necessary for preserving lung morphology during cryosectioning would likely disperse infiltrating immune cells in the airway lumen, making their individual study challenging. A complementary approach to address this pitfall could consist of collecting lavage fluid prior to inflation and cryosectioning to perform LC-MS/MS-based untargeted lipidomics and cell differential counts to characterize lipidomic changes correlated with immune cells. Importantly, we did not collect lavage fluid prior to sectioning in order to preserve tissue section quality for MSI acquisition. We have added an additional sentence explaining this future direction within the discussion of the main text. To distinguish polyunsaturated PEs differences across eosinophils, neutrophils or related cytokines is certainly desirable, but not achievable with today's methods.

Rev2, Comment 6: As stated in line 301, "displayed distributions that appeared to be independent of potential effects induced by treatment or sex", In figure S4 corresponds to Figure 3, all samples are shown, whereas large variance seemed to exist among individual samples. Detailed information of individual samples including sex and ozone exposure should be provided. And the impact of ozone exposure on these three lipids should be discussed.

Response to Rev2, Comment 6: Lung sections in **Figure S4 and Figure S5** are now numbered according to acquisition order, and information for each sample can be found in **Table S3**. The figure legends in both the main text and supplemental material have been updated to clarify information regarding each image shown. We displayed the spatial distributions of PE 18:0/22:6, PI 36:4, and PA 16:0/16:0 as each of these lipids were localized to morphologically relevant lung regions validated by our H&E staining. We did observe significantly altered abundance of these 3 lipids in the alveolar epithelium of lobes derived from HDM + O₃ exposed female mice. However, the regional distribution of each of these lipids did not change with respect to exposure.

Rev2, Comment 7: In Figure 3, are PE 18:0/22:6, PI 36:4, and PA 16:0/16:0 the top 3 distinct lipids? and how are they selected from all annotated lipids as examples? Why is not PI 36:4 annotated at the acyl chain level as the other two?

Response to Rev2, Comment 7: We observed other lipid species that were colocalized with these 3 lipids in negative ionization mode, which are included in **Table S4**. We chose these species as they were consistently detected across each lung section and did not exhibit changes in spatial distribution following combined HDM + O₃ exposure. Therefore, these lipids were representative of specific clusters annotated by our segmentation results that could be used as potential markers to distinguish individual lung regions. PI 36:4 was not annotated with acyl chain information as the peak matched to PI 36:4 in the reference

library used in our LC-MS/MS validation did not have this information available. Unfortunately, detailed acyl-chain information (e.g. PI 20:4_16:0 as a possible molecular species for PI 36:4) cannot always be readily asserted from MS/MS spectra. New instrumentation such as the Sciex ZenoTOF provides better fragmentation (including to deduce double bond positions), but this instrument is not available (yet) for MS-imaging studies.

Rev2, Comment 8: Which cluster in Figure 4A(cluster1-11) do pixels in Figure 4B-D correspond to? the information should be labeled on figures or in legends.

Response to Rev2, Comment 8: We have now added the specific clusters corresponding to each region in the **Figure 4** legend.

Rev2, Comment 9: Comparing Fig. 5B-D to 5A, it seems that MS signals exist in the airway lumen, otherwise the signal may be attributed to a much thinner lumen cavity in the sections of Fig. 5B-D compared to 5A. The authors should state whether the sections are consecutive and explain the reason underlying the variation.

Response to Rev2, Comment 9: The sections acquired in **Figure 5** are consecutive and were cut at 15 μm thick in both the H&E stained and MSI section, which has been specified in the figure legend. Section thickness was optimized for cryosectioning to prevent folding, tearing, and scoring of lung tissue at thicknesses less than 15 μm that were tested during method development, which we have clarified in the methods section (Line 174). Consequently, we could not directly superimpose our MSI ion images with our H&E-stained sections due to morphological variations between sections taken 15 μm apart. Nonetheless, we were still able to draw comparisons between our histological data and segmented MSI data in **Figure 5** since we could visualize clearly distinguishable lung regions in both sets of images.

Reviewer #3 (Remarks to the Author):

In this manuscript, the authors provided a framework for future mass spectrometry imaging experiments capable of relative quantification across biological replicates and expansion to multiple sample types. It was a great exploration and worth publishing in Nature Communications. Before the manuscript is published, I still have the following concerns.

Rev3, Comment 1: As in your manuscript, following MALDI-TOF MS data acquisition, matrix-coated tissue slides were scraped, and lipids were extracted for lipidomics. Do you consider the effect of MALDI matrix for MS detection?

Response to Rev3, Comment 1: We did not compare LC-MS/MS data acquisition between tissue slides coated with and without MALDI matrix. However, previous reports suggest that lipid detection is not significantly impacted by matrix deposition (doi: 10.1021/acs.analchem.3c05850). We have now added this reference to the main text and added a sentence providing additional information regarding our lipidomics data acquisition (Line 312).

Rev3, Comment 2: The choice of MALDI matrix is a critical factor for lipid detection. Did you evaluate different matrices for lipid detection in your study?

Response to Rev3, Comment 2: We acquired test sections with dihydroxybenzoic acid (DHB), daminonaphthalene (DAN), and 9-aminoacridine and selected matrices with the broadest lipidome coverage and signal in each ionization mode. Additionally, DHB and DAN are widely accepted matrices for use in positive and negative mode, respectively (doi: 10.1002/jms.4491). We have now provided brief justification for our matrix choices in the method section.

Rev3, Comment 3: You applied sparse LOESS normalization to minimize systematic errors while preserving biological differences in the spatial distributions of each compound. LOESS normalization is commonly used in LC-MS/MS-based metabolomics assays. Could you elaborate on why LOESS normalization is advantageous in MSI data, especially when addressing variability within and across samples? Additionally, how does LOESS compare to other common normalization methods in terms of its specific strengths and limitations for MSI?

Response to Rev3, Comment 3: Great question, and we added the following explanation into the method section. By using LOESS normalization in MSI data, we were able to account for gradual changes in signal intensity that are common in long experimental runs in MSI experiments. Specifically, the average acquisition time for a single lung section in our experiment was approximately 2.5 hours. Over the course of the imaging run, the laser intensity gradually decreases unless it is progressively adjusted during acquisition. Our implementation of LOESS in MSI was particularly advantageous for addressing this systematic error. Additionally, LOESS normalization was applied to each annotated compound individually. This is an important distinction from conventional TIC and root-mean square (RMS) normalization in MSI, which considers all compounds as a whole before applying normalization to each compound. This skews the normalized intensities if artifacts are still present in the dataset after filtering, which is in contrast to LOESS that can be applied to each compound separately. Lastly, compound-to-compound normalization consumes fewer computational resources than applying normalization to an entire dataset at once, which enables this method to be scaled with increasingly large MSI experiments. We have updated our RegioMSI package with options for performing conventional MSI normalization to enable comparisons to sparse LOESS normalization.

Rev3, Comment 4: You mentioned that the use of agarose inflation and embedding preserved the cellular integrity of lung tissue and avoided ion suppression and detector contamination caused by OCT. However, the degree of separation between the airway basement membrane and distal airway epithelium varied among samples. Are there any additional optimization steps that could be taken to improve the consistency of this separation, especially when applying the method to other more complex or fragile tissue types?

Response to Rev3, Comment 4: While agarose is cryoprotective, prevents detector contamination, and enables detection of lipids in MSI, OCT or fixed and embedded tissues retain greater morphological detail compared to agarose inflated and embedded sections. Further optimization of the inflation medium itself may be difficult, but including technical replicates of the same lobe and varying the sectioned location of the tissue block may enable more consistent visualization of the airway epithelium and basement membrane. Unfortunately, we were unable to image multiple sections from the same tissue block in this study as additional sections were used for optimization of sample collection, processing and acquisition. Additional experiments using the analytical workflow presented in this study could include both technical and biological replicates across a larger sample size to comprehensively characterize adjacent lung regions across all samples. We now acknowledge the need for optimization of sample preparation in the discussion section within the main text.

Rev3, Comment 5: Can mass spectrometry imaging distinguish the specific locations of double bonds in lipids? Did you consider double bond positions in the development of this method?

Response to Rev3, Comment 5: Unfortunately, mass spectrometry imaging solely using the time-of-flight detector present in the instrument used to acquire the samples in our study is not sufficient for distinguishing lipid double bond position. Acquiring samples using a combination of trapped ion mobility spectrometry and time-of-flight mass spectrometry could in principle obtain this information. However, this capability was not available for our experiment. We have now added a sentence to the discussion explaining this study limitation. Other instruments, like the Sciex ZenoTOF mass spectrometer, can detail double bond positions but are not available for MS-imaging.

Rev3, Comment 6: Does mass spectroscopic imaging require more regional selection of tissue samples? In this experiment, how to ensure the consistency of the selection of lung tissue regions?

Response to Rev3, Comment 6: Our unsupervised clustering approach was able to consistently identify major lung regions in all study samples despite large variation in tissue morphology from sample to sample. As the reviewer noted, the consistency of selecting regions of interest will certainly be improved in future method developments that were not available when we performed our study. For example, we are currently developing an updated clustering method that considers all peaks in an MSI dataset and all samples, which builds upon the method outlined in this study that individually segments samples using only annotated peaks. This updated method may be able to establish marker lipids for individual lung regions that could be subsequently implemented in a supervised learning algorithm to consistently select lung regions across samples. This method will be released in a future update of RegioMSI and could be coupled with spatial transcriptomics for cross-validating selected regions in MSI data. We have added a small comment on this issue in the method section.

Rev3, Comment 7: In this experimental model, sphingolipid and glycerophospholipid abundance showed differences in male and female mice. Have there been relevant reports before? Do your results have implications for the construction of such lung disease models in the future?

Response to Rev3, Comment 7: Our previous work cited within the main text as well as others have reported lipid class changes in male and female mice (Stevens et. al. 2023, doi: 10.1093/toxsci/kfab081, and doi: 10.1164/rccm.201508-1599oc). Our study here identifies lipid class changes at high spatial resolution, which is novel and important for understanding the molecular mechanisms of ozone induced exacerbations in asthma. Ozone elicits region-specific toxicity within the lung, suggesting that individual cell types may be responsible for producing responses that result in adverse effects as a result of exposure. The region-specificity of ozone and multitude of heterogeneously distributed cell types within the lung necessitate the detailed spatial characterization conducted in our study. Combining MSI with orthogonal techniques such as spatial transcriptomics and scRNA-Seq could inform detailed mechanistic studies spanning multiple conditions to establish models of ozone-induced allergic asthma.

We thank the reviewers for their additional comments and suggested edits to our revised manuscript. We have addressed these final remarks regarding our manuscript accordingly.

Reviewer 2:

Rev2, Comment 1. In the last paragraph of the Methods section, on line 234, "as previously" should be changed to "as previously described."

Response to Rev2, Comment 1: We have made the requested change to the manuscript.

Rev2, Comment 2. The labeling in Figure 6 should be enlarged and simplified for better visualization. For better understanding, the lipid classes should be labeled in the same form as mentioned in the main text, e.g., PE, PS, with abbreviations included in the legends.

Response to Rev2, Comment 2: The axis and legend label text for Figure 6 has been enlarged to enhance readability. Additionally, we have abbreviated lipid class names in panels A and C as requested, which are defined in the figure legend.

Rev2, Comment 3. The attached code appears to be of sufficient quality and accessibility, and it functions correctly with the demo data for two samples. However, since the demonstration of the RegioMSI package includes only a subset of the data, as noted in the Reporting Summary file, it is not possible to fully evaluate whether the results presented in the manuscript are reproducible. In addition, the object 'd' in the peak processing code is not clearly defined or explained, which may cause confusion.

Response to Rev2, Comment 3: We have described "d" in the RegioMSI package tutorial to clarify this object's purpose within the tutorial. To adhere to community standards regarding R package development and to limit the user hard disk space required by our package, we could only include a subset of data in our source code as the total size of our MSI data for both ionization modes exceeds ~100 GB. Alternatively, we have deposited the raw .imzml and .ibd files in Zenodo under the accession code 10.5281/zenodo.14846221, which can be downloaded and processed using our RegioMSI package.